# Gene-Based Resistance to *Erysiphe* Species Causing Powdery Mildew Disease in Peas (*Pisum sativum* L.)

**DOI:** 10.3390/genes13020316

**Published:** 2022-02-08

**Authors:** Jyoti Devi, Gyan P. Mishra, Vidya Sagar, Vineet Kaswan, Rakesh K. Dubey, Prabhakar M. Singh, Shyam K. Sharma, Tusar K. Behera

**Affiliations:** 1ICAR-Indian Institute of Vegetable Research, Post Box 1, Jakhini, Varanasi 221305, India; jyoti17iivr@gmail.com (J.D.); Vidya.Sagar1@icar.gov.in (V.S.); rksdubey@gmail.com (R.K.D.); pmsiivr@gmail.com (P.M.S.); 2ICAR-Indian Agricultural Research Institute, Pusa, New Delhi 110012, India; gyan.gene@gmail.com; 3Department of Biotechnology, College of Basic Science and Humanities, Sardar Krushinagar Dantiwada Agricultural University, Palanpur, Gujarat 385506, India; vineetkaswan@gmail.com; 4CSIR-Institute of Himalayan Bioresource Technology, Palampur 176061, India; skspbg@yahoo.co.in

**Keywords:** *er* gene, *Erysiphe*, marker-assisted selection, *Pisum*, powdery mildew

## Abstract

Globally powdery mildew (PM) is one of the major diseases of the pea caused by *Erysiphe pisi.* Besides, two other species *viz*. *Erysiphe trifolii* and *Erysiphe baeumleri* have also been identified to infect the pea plant. To date, three resistant genes, namely *er1*, *er2* and *Er3* located on linkage groups VI, III and IV respectively were identified. Studies have shown the *er1* gene to be a *Pisum sativum Mildew resistance Locus* ‘*O*’ homologue and subsequent analysis has identified eleven alleles namely *er1–1* to *er1–11*. Despite reports mentioning the breakdown of *er1* gene-mediated PM resistance by *E. pisi* and *E. trifolii,* it is still the most widely deployed gene in PM resistance breeding programmes across the world. Several linked DNA markers have been reported in different mapping populations with varying linkage distances and effectiveness, which were used by breeders to develop PM-resistant pea cultivars through marker assisted selection. This review summarizes the genetics of PM resistance and its mechanism, allelic variations of the *er* gene, marker linkage and future strategies to exploit this information for targeted PM resistance breeding in *Pisum*.

## 1. Introduction

Globally, the pea (*Pisum sativum* L.; 2n = 2x = 14) is one of the most important cool season legumes consumed both as a vegetable and as a pulse. The pea genome is estimated to be 4.45 Gb, making it one of the largest among the legumes [1]. Peas are low in fat but high in fiber, protein, vitamin C, ß-carotene, thiamine, riboflavin and iron content, thereby making it a healthy food capable of meeting the global dietary needs of over 900 million undernourished people [2]. The rich genetic diversity of *Pisum* has helped this crop to cover the vast geographical area under cultivation [3]. Global area and production of green peas have nearly doubled during the last two decades from 1999 (1.5 mh; 11.39 mt) to 2019 (2.8 mh; 21.76 mt), respectively. However, only a slight increase in productivity has been recorded rising from 7.6 t/h in 1999 to 7.8 t/h in 2019 [4]. Even those Asian countries with a very high production showed a similar trend, with a minor increase in productivity from 8.3 to 8.5 t/h from 1999 to 2019. This indicates that despite all efforts to increase the yield, the biotic and abiotic stresses continue to play a significant role in yield reduction [5].

Pea productivity is constrained by numerous fungal pathogens of which powdery mildew (PM) caused by *Erysiphe* species (order Erysiphales, family Erysiphaceae) is the major one. Although the order Erysiphales includes nearly 19–22 accepted genera and 400 species, the majority of pathogenic species belong to the genus *Erysiphe* [6]. These obligate biotrophic parasites infect nearly 10,000 species of angiosperms, including cereals, pulses, fruits, vegetables and ornamental plants [7,8]. Many of these are host-specific or target a very small number of hosts, suggesting the presence of very specific pathogenesis-related (PR) genes. The pathogen infection on plants is distinguished by easily recognizable patches of white to greyish talcum-like growth with its different causal organisms in various vegetable crops, including peas (*Erysiphe* spp.) [9]. 

The PM can cause up to 25–70% yield losses in peas with a negative impact on yield and its contributing traits (Figure 1) [10]. When the crop is grown for industry processing or seed purposes, the problem becomes more severe, especially under warm dry days and cool-night climatic conditions [11]. The pathogen has been reported all over the world, most notably in pea-growing countries like India, Pakistan, China, the United States, Russia, Germany, the United Kingdom, Italy and Ethiopia. The air-borne movement of the pathogen spores and sexual recombination aid in the production of new virulent races, allow for the rapid dissemination and adaptability of this devastating pathogen [12]. Although PMs are the most prevalent plant pathogenic fungi, detailed research into the management of this disease is limited due to its obligatory biotrophic nature, which makes ex-situ or in-vitro experiments difficult [13,14]. A few researchers have compiled the information about PM in pea, especially for pathogen control [15], marker-assisted breeding [16] and allelic variation at the *er* locus [17]. However, there is no comprehensive review covering the pathogen, novel variations of the *er* genes/alleles, advances in mapping strategies, linked markers and future strategies to combat the disease. With this backdrop, the current review encompasses various aspects of the host and pathogen and the recent strategies for the management of this deadly disease.

## 2. *Erysiphe* Species Infecting Peas

Although *Erysiphe pisi* is the most predominant fungal species causing PM in peas, still we could not precisely trace when this pathogen was first described as *E. pisi*. However, *Erysiphe polygoni* [18] and *Erysiphe communis* [19] were reported in the literature as early as 1925. In addition, two other species (*Erysiphe trifolii* and *Erysiphe baeumleri*) infecting peas have also been identified in the United States, Spain, India and the Czech Republic [15]. The first report of *E. baeumleri* infecting pea was in 2001 from the Czech Republic (North Moravia) on the ‘Highlight’ cultivar. The diagnostic traits of different samples of *E. baeumleri* and *E. pisi* were known. The symptoms caused by *E. baeumleri* progresses relatively slowly and appears mainly on the petioles or leaves and rarely on stems or pods [20].

Attanayake et al. [21] reported *E. trifolii* infection on peas and showed that this species can be distinguished from *E. pisi* using *rDNA* internal transcribed spacer (ITS) sequences and also through morphological characters of chasmothecial appendages that are primarily of the mycelioid type in *E. pisi* and dichotomously branched in *E. trifolii* and *E. baeumleri*. In addition, horizontally extended and colored appendages could easily distinguish *E. trifolii* from *E. baeumleri*. The virulence of these species varies based on the location and symptoms caused by *E. trifolii* and *E. baeumleri* being less severe than those caused by *E. pisi* in the USA and the Czech Republic, respectively. The *E. trifolii* expressed severe symptoms on the pea genotypes carrying *er1* gene at Kanpur, India [22].

## 3. Screening Methodology and Disease Scale

### 3.1. Screening under Natural Epiphytic Conditions

In any resistance breeding program, identifying a hot spot is crucial for the disease screening. The genotypes should be grown in replications with a sufficient plant population, along with the spreader rows of susceptible cultivars to ensure a uniform spread of PM inoculum [23]. Standard agronomic practices should be followed to ensure a healthy crop without the use of fungicidal spray. In the event of insufficient inoculum in the field, pre-maintained spores obtained from the susceptible plants should be used to dust over the testing population [24]. The reaction of the genotypes should be recorded 2–3 weeks after inoculation. The F_2_ plants can be used to study the genetics of the powdery mildew resistance in the F_2:3_ generation. The trials must be conducted in replication for more reliable conclusions [24].

### 3.2. Controlled Conditions: Detached Leaf Method

The detached leaf assay has been extensively used for better screening of the genotypes to PM resistance under controlled conditions [25,26,27,28,29]. For this assay, 15–30 days old plant leaflets with petioles are first floated in the Petri dishes containing 5% sucrose and benzimidazole (40–50 ppm) to improve the longevity of detached leaflets. These leaflets are then dusted with PM inoculum, with a camel hair brush or a paintbrush followed by incubation at 25 ± 1 °C under 16/8 h day/night cycle [28,30]. The observations should be recorded as per the symptom development, which can occur between 3 to 14 days after inoculation [25,31]. 

### 3.3. PM Disease Scale in Pea

Different PM disease scoring scales have been proposed by different researchers and the scales of 0–4, 0–5, 1–5, 0–9, and 0–10 have been used for scoring the PM disease (Table 1).

## 4. Genetics and Resistance Mechanism 

### 4.1. Genetics

The most economical means of any disease management could be to incorporate resistance gene(s) into any promising commercial varieties lacking such gene(s) [15]. Studies on the inheritance of PM disease revealed three genes, namely *er1, er2* and *Er3*, conferring resistance to *Erysiphe* species [22], (Figure 2). An investigation into the genetics of powdery mildew resistance (PMR) has been carried out in the past, which has shown different modes of inheritance including single recessive [35,36], single dominant [9,37] and duplicate recessive gene actions [38,39] (Table 2). Hammarlund [19] was the first to investigate PM resistance in peas and reported cumulative factors for susceptibility. Harland [36] discovered some resistant plants in a local Peruvian variety, wherein a single recessive gene (*er*) control has been recorded, which is now known as *er1.* The possibility of the multigenic nature of PM had also been supported by many working groups [26,27,28,40,41,42]. However, it is well-known that the recessive *er1* gene is responsible for the majority of naturally occurring PM resistance [43,44,45,46] including the two induced recessive mutations, *er1mut1* and *er1mut2* [47].

The third resistance gene (*Er*3) was identified from the wild relative *Pisum fulvum* line ‘P660-4’ (a selection from ICARDA accession ‘IFPI3261’ from Idlib, Syria), which showed dominant gene action for resistance [9]. *Er*3 was then introgressed in the cultivated genotypes through hybridization (via male parent) and is now available for use in the breeding program. Recently, Bobkov and Selikhova [37] have confirmed the presence of another dominant gene *(*identity yet not confirmed) in the *P. fulvum* line (i-609881), which was originally collected from UIP (Saint Petersburg, Russia). This gene is also being successfully introgressed into various cultivated genotypes through repeated backcrossing.

### 4.2. Resistance Mechanism and the Temperature-Based Reaction of Resistant Genes

The resistance mechanism of genes imparting PM resistance have also been studied at the cellular level [27,48] and *er1* was found to impart resistance by inhibiting *E. pisi* invasion of pea epidermal cells. In most pea accessions carrying the *er1* gene, the vast majority of *E. pisi* conidia germinate and form appressoria, but with restricted pathogen growth and no secondary hyphae formation [48]. In contrast, the *er2* gene mediated resistance is based mainly on post penetration cell death, mediated by a hypersensitive response (HR). However, on *Er3* genes carrying plants, most of the *E. pisi* conidia penetrates pea epidermal cells and form secondary hyphae, but growth of these established colonies is prevented by a strong HR [28,49], Figure 2. The defense mechanism of HR involves accumulation of reactive oxygen species (ROS), antimicrobial proteins and phytoalexins [48,50,51].

The *er1* reportedly provides moderate to complete resistance to all plant parts, whereas *er2* (JI2480) confers PM resistance only to the leaves (tissue specific) and is also influenced by leaf age and temperature. Furthermore, penetration resistance conferred by the *MLO* mutations were associated with formation of papillae in the penetration sites. These papillae are created primarily by the deposition of callose matrix comprising inorganic and organic compounds, which function as physical or chemical barriers to pathogen penetration [48,50]. Besides, formation of protein cross-linking in the host cell wall hampering haustorium formation is also found responsible for resistance mechanisms [48].

Although detailed studies for *er1*-based resistance and its temperature-independent responses are known [27] (Table 3), *er2* and *Er3* genes are not yet properly characterized for the temperature response. In the case of *E. pisi*, the *er2* genotype (JI2480) showed complete resistance at 25 °C while incomplete resistance at 20 °C and susceptibility at 15 °C [27]. In contrast, this line was completely resistant against *E. trifolii* at 20 and 25 °C [22]. These results suggest that the resistance in JI2480 to *E. pisi* is temperature-dependent, while the interaction between this genotype and *E. trifolii* is temperature-independent. The researchers further demonstrated that *E. trifolii* could overcome *er1* and *Er3* resistance in some conditions. However, *er2* demonstrated very high resistance to *E.*
*trifolii* under all environments including locations. It has also been discovered that the *er2* gene is effective against both *E. pisi* and *E. trifolii* [22].

In addition, *er2* was found in a few resistant pea accessions only *viz.*, SVP 950 [18], SVP-750, SVP-951, SVP-952 [18,55] and JI 2480 [30], which was subsequently transferred in the different background of pulse and vegetable types.

There have been reports of a breakdown of the *er1* by *E. pisi* [56,57] as well by *E. trifolii* under controlled and field conditions [22] (see Figure 2). However, *Er3* is found completely effective against the *E. pisi* and may also be effective against *E. trifolii* in the regions where the growing temperature does not typically reach 25 °C or above. Some of the various *er1* alleles that have been reported, such as *er1-1* and *er1-2*, are currently used in pea PM resistance breeding programs in China [58,59].

## 5. Biochemical and Molecular basis of PM Resistance

### 5.1. The Biochemical Aspect

In response to PM infection, pea plants undergo a series of anatomical, morphological, physiological, biochemical and molecular changes. The resistant (*R*) genes present in the plant work in tandem with the defense mechanism operational against PM fungal infection. In a recent review, Martins et al. [60] have comprehensively highlighted the multi-layered array of PM defense mechanisms in various legumes. The complex PM infection response results in the rapid generation of reactive oxygen species (ROS), which include free radicals such as superoxidase anion (O_2_^−^), hydroxyl radical (OH), and nonradical molecules like hydrogen peroxide (H_2_O_2_) and singlet oxygen (^1^O_2_) (Figure 3). The abundance of ROS eventually leads to increased oxidative damage and ultimately cell death [61]. Thus detoxification of excess ROS could be achieved by an efficient enzymic antioxidant system (*viz*., superoxide dismutase (SOD), catalase (CAT), guaiacol peroxidase (GPX), enzymes of ascorbate-glutathione (AsA-GSH) cycle such as ascorbate peroxidase (APX), monodehydroascorbate reductase (MDHAR), dehydroascorbate reductase (DHAR) and glutathione reductase (GR)) as well as non-enzymatic antioxidants (Ascorbate (AsA), glutathione (GSH), carotenoids, tocopherols and phenolics) [61].

Many studies have suggested that various biochemical parameters play a role in PM disease resistance in different pea genotypes. The role of phenolic compounds in induced resistance to PM infections was initially demonstrated by Maranon [62]. Additionally, the role of different biochemicals was identified in controlling PM resistance such as high phenols and proteins [63]; phenols and peroxidase [64]; alkaloids, proteins, proline, polyphenol oxidase, and peroxidase [65]; peroxidase, polyphenol oxidase and total phenols [66]; total phenol, proteins, polyphenol oxidase, peroxidase, chitinase, and β-1,3-glucanase [67]; SOD and CAT activity [68]. As a result, these bio-markers could be used to identify the resistant plants in the early stage of PM resistance breeding. 

On contrary, the sugar content was found higher in the susceptible pea genotypes [63]. When compared with the crops like wheat [12,70], brassica [71], and grapes [72], PM in *Pisum* is still a poorly investigated trait in terms of defense-related secondary metabolites and their protein products.

Proteomic analysis of a PM-resistant pea genotype JI2480 (carrying *er2* gene) and a susceptible cv. Messire (under control and infected conditions) revealed more defense-related proteins accumulation in JI2480 than Messire, which mainly belongs to three functional categories, *viz*., photosynthesis, carbohydrate catabolism and stress related responses [69]. In addition to the pea as a host, some proteomic studies on the pathogen have also been conducted. Noir et al. [73] presented the first functionally annotated proteome of a PM fungus infecting barley using 2D gel electrophoresis with MALDI-TOF MS and MALDI-TOF/TOF MS/MS. A total of 123 distinct proteins belonging to different metabolic pathways such as lipid, carbohydrates, proteins, and protein processing were identified, which indicate that the protein machinery of conidia is required for meeting the needs as storage structure and germination processes for pathogen multiplication. The proteomic studies of *E. pisi* isolates showed a high proportion of protein-machinery and heat shock proteins (HSP). The HSPs are a vital component of cell regulatory machinery and play an important role in the survival and spread of the biotrophic *Erysiphe* pathogen [14].

H_SP_90 is required not only for pathogen survival, but also for thermal transitions during the growth cycle, which maintains cellular adaptations [74]. This protein is controlled at transcription and post-transcriptional levels following a heat shock [74]. H_SP_90 plays critical roles in the folding and maintenance of a subset of proteins known as client proteins like phosphatases and kinases. A H_SP_90 client protein, MAP-Kinase, is an essential component of the cell integrity signaling pathway, which activates the transcription factors required for cell wall integrity maintenance [74]. During the transcriptomic study of the *E. pisi;* the protein kinases, phosphatases, HSPs and ATPase were identified as the putative effector, with a role in the pathogenicity and virulence [75]. Arthur et al. [76] suggested that many *R* genes require highly conserved chaperone molecules to limit the pathogen growth. In case of peas, the resistant genotypes were reportedly having two Hsp90 homologues which may contribute to regulate powdery mildew resistance in garden pea [75].

### 5.2. The Molecular Aspect

The pea plant protects themselves against any fungal invasion by activating a set of defense response genes as studied widely in different plant species like *Arabidopsis* [73,74,77], *Medicago* [78], barley [76,79,80,81], and peas [82]. Barilli et al. [82] studied the gene expression profile of PM-infected and healthy plants (24, 48 and 72 h after inoculation) in three different genotypes of peas JI2302, JI2480 and IFPI3260 carrying *er1*, *er2* and *Er3* genes, respectively. Furthermore, of 20 studied genes, 16 showed differential expression. Induction of *Chi2* gene that encodes an endo chitinase enzyme responsible for antifungal activity in the resistant genotypes (JI2302 and JI2480) after the PM infection was recorded. Similarly, the *Prx7* (encoding an elicitor-inducible peroxidase) expression also got significantly induced after PM infection in the resistant lines JI2302 and IFPI3260. Contrary to this, *Prx7* was found to be down-regulated in infected leaves of JI2480 (*er2*) at 48 and 72.0 h after inoculation. Conclusively, leaves of JI2302 (*er1*) showed mainly *Pschitin* and *Chi2* as well as genes encoding for pea defensins, whereas leaves of IFPI3260 (*Er3*) showed the highest expression of *DRR230a*, *DRR230b* and *DRR230c* (encoding pea defensins) and *Prx7* after pathogen inoculation. Compared to *er1* and *Er3* genotypes, JI2480 (*er2*) also showed *Pschitin* and *Chi2* accumulation, but with reduced activation of pea defensins.

During infection the *E. pisi* (*Ep*) secrets, a number of effectors through haustoria thereby establishes itself in the host. Studies have identified a number of candidate effector proteins, which can be used to manage the PM in the peas [83]. The RNA-Seq analysis of *Ep*-infected pea leaves have identified the candidate-secreted proteins (CSPs) and the candidate-secreted effector proteins (CSEPs) [83]. The qRT-PCR of a few *Ep*CSEP/CSPs confirmed their infection-stage-specific expression and also expression in the haustoria. Host-induced gene silencing has also established the functional roles of *Ep*CSEP001, *Ep*CSEP009 and *Ep*CSP083 genes, while foliar application of *Ep*CSEP/CSP dsRNAs showed a great reduction in the PM disease expression. Homology studies showed the analogous nature of *Ep*CSEP001 and *Ep*CSEP009 with that of fungal ribonucleases belonging to the RALPH family of effectors [83]. RNA seq analysis of *E. pisi*-infected resistant (JI-2480) and susceptible (Arkel) genotypes showed glycolysis as the key energy source pathway during infection. Moreover, transcription factors like-WRKY-28 and a number of putative pattern recognition receptors, were observed differentially regulated in the resistant genotype, which indicated the activation of host-mediated defense responses when infected with *E. pisi*. Additionally, in-silico effector search have also identified various putative effectors like peptidyl-prolyl cis-trans isomerase or cyclophilin (CYP) [75].

## 6. Molecular Characterization of *er* Genes on Linkage Groups

### 6.1. Allelic Variations at er1 Locus

The genetic basis of *er1* resistance was first reported as a monogenic recessive nearly 73 years ago [36] and is still very stable and effective gene imparting PM resistance in peas. Afterwards, several new alleles have been identified that were derived from either natural or artificial mutagenesis in the PM susceptibility gene, which is part of the mildew resistance locus ‘O’ (*MLO*) gene family (*PsMLO1*) [84,85]. This was also supported by Bai et al. [86], who reported that *er1* and *mlo* resistance share common genetic and phytopathological features. Furthermore, resistance conditioned by *mlo* alleles was observed to function early and typically terminate the pathogenesis before the fungus invades the first host cell [87]. Such a type of immunity was initially reported in a mutant barley population and also in an Ethiopian landrace [88].

Humphry et al. [84] found that the resistance in pea lines JI210, JI1559, JI1951 and JI2302 was due to loss of function in the *PsMLO1* locus; whereas resistance in JI2480 (carrying *er2*) line is caused by a failure of a different gene. Similarly, resistance mediated by the *Er3* gene is unrelated to *PsMLO1* because both the genes were located on different linkage groups (LGs). Under field conditions, the *mlo*-based (null allele) PM resistance in barley was found to be complete, while the pea genotypes JI210, JI1559, JI1951, and JI2302 with a null mutation at *PsMLO1* showed incomplete resistance to PM. Thus, in pea other *MLO* homologues may also be contributing to the PM susceptibility [84].

The *er1* gene is known to encode a *MLO1* like transmembrane protein with a calmodulin-binding domain. Calmodulin is a calcium binding protein where calcium usually acts as an important messenger of stress. The presence of a calmodulin-binding domain indicates the role of the *er1* gene in disease response signaling by perceiving the stress signal at the cellular membrane. The structure of the barley *MLO1* protein shows that it is composed of seven transmembrane helices and that mutations in cytoplasmic and transmembrane domains of the protein result in impairment of function, leading to disease resistance (https://www.uniprot.org, accessed on 25 September 2021). Similar structural information is still missing for proteins encoded by *Pisum er* genes and alleles. 

To date, 11 alleles of *er1* have been identified that include *er1-1* to *er1-11,* which represent prevailing variants of the *er1* gene in resistance sources from different geographical origins (Table 4, Figure 2). In an induced mutation study in pea (using ethyl nitrosourea), Leitão and coworkers [47] have identified two altered genes and were named as *er1mut1* and *er1mut2* in genotypes Solara and Frilene, respectively. Later, the same group could succeed in the sequence-based characterization of the identified mutation and the S(*er1mut1)* was found having C/G transversion in exon 6, while F(*er1mut2*) was due to the G/A transition in exon 10 [89]. Complete co-segregation of the KASPar marker KASPar-*er1-1* with the known sequence tagged site (STS) functional marker *er1-1*_S (*er1mut1*)_STS, was consistent with the identity of S(*er1mut1*) as *er1-1* [90]. In addition, Ma and coworkers [90] have also described the induced mutation ‘F(*er1mut2*)’ as *er1-10*; while Sun and coworkers [91] have also mentioned ‘S(*er1mut1*)’ and ‘F(*er1mut2*)’ as *er1-1* and *er1-10,* respectively. 

Humphry et al. [84] and Pavan et al. [92] reported five *er1* resistant alleles *viz., er1-1* (JI1559), *er1-2* (JI2302), *er1-3* (JI210), *er1-4* (JI1951) and *er1-5* (ROI3/02). Subsequently, Sudheesh [93] reported a 2-bp insertion in intron 14 in the resistant lines ‘Yarrum and ps1771’. This variation was later described as *er1-11* allele [90]. During 2016, two other novel alleles *viz., er1-6* (G0001778; [59]) and *er1-7* (DDR-11; [58]) were also characterized. In 2019, *er1-8* (G0004839) and *er1-9* (G0004400) were discovered, which were characterized by a 3-bp (GTG) and a 1-bp (T) deletion in the wild-type *PsMLO1* gene, respectively. Many reports clearly demonstrated that the *er1-1* and *er1-2* are the most common variations at the *er* locus, and Chinese accessions are far more characterized than any other accessions worldwide [53,94]. Interestingly, Sun et al. [91] studied 55 accessions and found that Chinese accessions (15 accessions carrying *er1-1, er1-2*, *er1-6* and *er1-7*) has the highest allelic diversity at the *er1* locus, followed by the USA (13 accessions; *er-1-2* and *er1-6*) and Australian accessions (6 accessions; *er1-1*, *er1-2*, *er1-9*). On the contrary, in a set of Indian accessions held at ICARISAT, Hyderabad, only the *er1-2* variant was detected [91]. 

### 6.2. Linkage Groups (LGs) of er1, er2 and Er3

The linkage of the *er* gene with the morphological marker “Gritty” (*Gty*) was observed and both the factors were assigned to the LGIII [97]. Subsequently, Wolko and Weeden [98] have placed the *Gty* gene on LGVI. However, with the advancements in molecular breeding tools, the position of these genes is now confirmed (Table 5, Figure 2). The detailed studies have placed the *er1* gene on LGVI [99,100], while the *er2* gene was localized on LGIII [28]. Fondevilla et al. [101] initially mapped the *Er3* gene between the SCAR marker ‘Scw4637’ and the RAPD marker ‘OPAG05 1240’ on an unknown pea LG. However, Cobos et al. [102] have confirmed that two markers *viz*., AA349 and AD61 were linked to the *Er3* gene which was located on the LGIV at 0.39 cM downstream of marker AD61.

### 6.3. Comparative Mapping

In addition to three genes contributing resistance to PM, there have been a few reports of uncharacterized and incomplete resistance against *Erysiphe* spp. in peas [9,15]. The *MLO* locus has been associated with susceptibility in several plant species including legumes. Santos et al. [105] used genetic maps from *Lathyrus sativus* and *L. cicero*, as well as genome from *P. sativum*, *L. culinaris*, and *M. truncatula* to develop a comparative linkage map of the *MLO* locus. This map was constructed with the aim of gaining information about the synteny, conserved sequences of the *MLO* locus and chromosomal arrangements that exist among these legume species. The *LsMLO1* is located on the upper part of LGI and is macrosyntenic to the *P. sativum* chr1LG6. The locations of *MLO1* in *P. sativum*, *L. culinaris*, and *M. truncatula* are at chr1LG6, chromosome 2, and chromosome 6 respectively [106,107], all of which are syntenic to *L. sativus* LGI. However, the microsynteny between the adjacent markers were not detected between the *L. sativus* linkage map and the *P. sativum* genome. 

## 7. Breeding for Powdery Mildew Resistance

### 7.1. Conventional Approaches

Plant breeders have identified several sources of PM resistance following the screening of large collections of pea germplasm (Table 6). In these identified genotypes, the resistance is primarily controlled by monogenic factors, thus eliminating the effect of the environment on the expression of genes related to PM resistance [108]. If additional factors such as quantitative loci were to be considered, the role of environment in governing resistance could not be ruled out while breeding for PM resistance [15,30]. The knowledge about the role of a favorable environment in disease development has been applied to the cultivation of otherwise susceptible cultivars that are not affected by the disease; this phenomenon is known as disease escape. For example, early-flowering group of garden pea cultivars, such as ‘Arkel’, which despite being susceptible to the disease [29], remains unaffected by the PM and does not suffer losses due to disease escape. The disease is reported to be more prevalent in late maturing or late planted varieties [10,109]. Thus, in addition to the genetic background, significant genotype × environment interaction for PM was observed in studies where PM infection was reported to be favored by long growth cycles [108]. This emphasizes the importance of multilocational/seasonal testing of the accessions for more reliable results [108]. Besides, germplasm augmentation and genetic enhancement including pre-breeding are to be in tandem and continuum of the breeding program.

Interestingly, most of the identified PM-resistant accessions (globally) were found to be carrying the *er1* gene. However, utilization of these sources is still a matter of concern, as many breeders lack the facilities for the precise screening and detailed characterization. Furthermore, many genotypes that reported PM resistance during the 1990s were found to carry undesirable traits like a poor yield, a low test seed weight (TSW) and susceptibility to lodging. However, there is now a wide variety of PM-resistant pea genotypes available, with a good yield potential, and a high TSW and lodging resistance [2].

### 7.2. Molecular Breeding Using Linked DNA Markers

Until recently, the *er1* gene was the most commonly used resource in pea breeding to develop PM-resistant cultivars. The cultivation of pea varieties with same PM resistance gene may result in the emergence of new pathogen race(s) following the breakdown of the resistance [15]. On the contrary, a combination of PM resistance genes and alleles may improve the resistance durability. Furthermore, due to the breakdown of resistance under varying environmental conditions, the only option left is to incorporate multi-gene resistance into the cultivated genotypes through gene pyramiding. However, due to overlapping phenotypes produced by the PM resistance genes, pyramiding through a traditional breeding approach is a strenuous exercise. In addition, handling an obligate pathogen like PM further complicates the selection process for PM resistance. To address these issues, molecular markers linked to the PM resistance genes may play a great role in identifying the resistance sources and also in the pyramiding of resistance genes in different pea genotypes. Several DNA markers linked to the PM-resistant genes (*er1*, *er2*, *Er3*) are known (Table 7), which are being used for the marker-assisted selection (MAS). Ghafoor and McPhee [16] provided an in-depth look at the potential of MAS for breeding PM resistance genotypes in peas. The mapping of PMR genes began in the 1990s, and most of these studies were done in F_2_ mapping populations using a bulked segregant analysis (BSA) approach with varying map distances. Later, these mapped regions were refined further to identify the closest possible markers linked to the identified genes. In addition, the DNA markers were also reported for various alleles of the *er1* gene (Table 8; Appendix A). The validation studies were also performed and markers like AD61, AD60, and *c5DNAmet* could be validated in different genetic backgrounds. Like the *er1* gene, focus is needed for the identification of allelic diversity of *er2* and *Er3* genes. Afterwards, allele-specific (AS) markers for *er2* and *Er3* should be developed so that the pea germplasms could be quickly and precisely screened using either KASPar or AS-PCR markers. This will ultimately help in the AS pyramiding of PM resistance genes (*viz*., *er1, er2* and *Er3*) in different cultivars (in different combinations) and then their precise deployment in the areas where large scale pea cultivation is being done.

## 8. Durable Resistance Strategies for PM Resistance

### 8.1. Gene Introgression from Related Species

From time to time, efforts have been made by the researchers to find out new sources of PM resistance, both in cultivated and wild *Pisum* accessions [9,49,118]. The majority of PM-resistant pea accessions were found to belong to the two subspecies *viz.*, *P. sativum* L. subsp. *sativum* and *P. sativum* subsp. *elatius* [119]. Interestingly, the majority of these accessions are carrying the *er1* gene. Another recessive resistance gene *er2* was discovered in a few resistant pea germplasm (mostly *P. sativum*) *viz.*, SVP-950, SVP-750, SVP-951, SVP-952 [18,55] and JI2480 [30]. Similarly, the resistant sources for the *Er3* gene have been identified in genotype of *P. fulvum viz.*, ‘P660-4’ [9,49]. However, as previously noted, there is a substantial need to search for resistant accessions in other related species, particularly with prior knowledge of crossability barriers of a primary (GP-1), secondary (GP-II) and tertiary (GP-III) gene pool of *Pisum*. There has been a lot of taxonomical debate about the species concept of *Pisum* [120]. The genus *Pisum* comprises one to five species, depending on taxonomic interpretation and the International Legume Database (ILDIS), and currently recognizes three species *viz*., (1). *Pisum abyssinicum* (syn. *P. sativum* subsp. *abyssinicum*); (2). *P. fulvum*; and (3). *P. sativum* with two subspecies viz., *P. sativum* subsp. *elatius* and *P. sativum* subsp. *sativum.* The primary gene pool includes the *sativum*/*elatius* complex, having nuclear-cytoplasmic incompatibility within the complex [121]. 

Fortunately, many studies on species hybridization barrier in *Pisum* have been conducted which resulted in the generation of F_1’_s, their quantitative characters *(*stem length, number of nodes, node of first flower, number of pods, seeds and seed weight) and fertility was analyzed [119,121,122,123,124,125,126]. These groups have adopted different approaches (evolutionary lineage concept of A, B, C and D) by keeping taxonomical distribution aside to study the species barrier. Bogdanova et al. [122] concluded that the divergent wild and endemic peas differ in hybrid sterility in reciprocal crosses from cultivated peas depending upon the allele of a nuclear speciation gene, ‘*Scs*1’ involved in nuclear-cytoplasmic compatibility. They reported highly sterile F_1_ displaying chlorophyll deficiency and variegation, reduction of leaflets and stipules when *P. sativum* subsp. *elatius* accession ‘VIR320’ was used as the female parent with domesticated peas (*P. sativum* subsp. *sativum*). On the contrary, reciprocal hybrids produced normal seeds [121]. This reflects the nuclear–cytoplasmic conflict/incompatibility within the *Pisum* subspecies. The cause of the phenomenon is not yet understood, but it could be due to altered metabolic processes in the plastids of F_1_ hybrids. *P. fulvum* is a wild *Pisum* species that has little hybridization success with *P. sativum* and is likely to suffer from linkage drag [127]. However, other researchers have successfully utilized this species to transfer the PM resistance into the cultivated genotypes [37,49,118].

### 8.2. Characterization and Introduction of Resistant Sources

For the development of an elite cultivar for a given ecosystem, the predicted resistance, durability and stability are some of the critical considerations in pea breeding programs [60]. Johnson [128] proposed a strategy for increasing the likelihood of attaining durable yellow rust resistance in wheat by utilizing a known parent cultivar with proven durable resistance. There are several resistant lines (Table 6) reported in *Pisum,* but most of these have not yet been adequately characterized in terms of the resistance mechanism at allelic level. Fortunately, it is easy to introduce a new cultivar with distinct *R* genes in a crop like *Pisum* because of the short growing cycle and ease in disease identification. Genetic evidence regarding the chromosomal position of PM resistance genes *er2* and *Er3* are known, yet cloning of these loci has not yet been reported. 

The preceding discussion has made it very clear that there is a pressing need to incorporate a wide range of genetic sources of resistance to PM in commercial pea cultivars. Unlike wheat, where nearly 200 resistant genes and several *QTL*s were known [12], there is a meager genic information known for PM resistance in pea. To date only three resistant genes have been reported, that too, with very limited testing for various isolates and species of PM fungus. Furthermore, of three PM resistance genes, use of *er2* and *Er3* is still very limited, with only a few reports of their introgression into some of the elite genotypes [9,28,37]. Although, *er1*, being recessive in nature, is giving a somewhat durable type of resistance to the pea genotypes against PM [27,59]. Still, the use of single gene-based resistance (which is also called as vertical resistance) approach by deploying only *er1* gene in most of the cultivated varieties, to control the PM pathogen is quite threatening and risky. Thus, to avoid any possible breakdown of *er1* based PM resistance, we must use all the available *er* genes through gene pyramiding approach in pea PM resistance breeding program. In addition, diverse PM resistance sources (varieties) of peas having various *er* genes and allelic combinations should be suitably deployed in the major pea growing areas (having diverse virulent PM races) to prevent the possible breakdown of any of the PM resistance gene(s). 

### 8.3. Gene Pyramiding and Crop/Cultivar Diversification

The concept of resistance gene pyramiding into a single cultivar through breeding is being advocated in many crops with considerable success [129,130]. More resistance genes should be identified and pyramiding of already known resistance genes should be attempted for the better management of this deadly disease, especially in the areas where individual resistance genes have already been exposed to the pathogen. The combination of all the three genes (*er1, er2* and *Er3*) could be stacked in one cultivar, which then provide a more stringent barrier to pathogen for the disease development by limiting their establishment (through *er1*) as well as triggering the hypersensitive response (through *er2* and *Er3*). This will then result in the potential increase in the durability of PM resistance in such genotypes [15]. Using linked molecular markers, these genes could be easily brought into a single cultivar [37], which is otherwise not possible through morphological selection as the genes are located on distinct positions on the genome and exhibit differential resistance expression. Efforts are being made to combine *er1* and *er2* genes in one background using MAS; however, the published literature on pyramiding of all the three genes is still scanty.

Furthermore, the resistance in the cultivated *Pisum* is very specific to a particular agro-climatic zone or environmental conditions, owing to the race-specific nature of PM resistance. Cultivar diversification is another approach for PM disease management that is primarily based on the availability of cultivars with known resistance genes that might be deployed in space or time or at the same time. For the management of the prevailing virulent race of any region, the resistant pea genotype carrying the most effective *er* allele/gene with superior agronomic traits can directly be recommended for any particular location. Moreover, identifying genotypes with resistance to multiple *Erysiphe* species and isolates may improve the durability of PM resistance. Some of the pea accessions *viz*., JI1559 and JI1951, had shown very high resistance to the multiple isolates of *E. pisi* [30] as well as to the *E. trifolii* [22]. Alternatively, these *er* alleles/genes can be rapidly transferred to other susceptible cultivars through marker-assisted breeding. Furthermore, overall PM disease pressure can be drastically reduced by undertaking the pea cultivation in the intercropping system, wherein non-host crop species can act as a physical barrier for the disease development and their spread. Villegas-Fernández et al. [131] have successfully demonstrated a clear reduction in the PM disease pressure in peas by intercropping of peas with barley and faba bean in a 50:50 ratio.

### 8.4. Utilization of Susceptibility (S) Genes and Gene Editing for Resistance

Loss of function mutations, such as *er1* and *er2*, result in recessive PM resistance [28,84,85,104]; the wildtype *Er1* and *Er2* genes are therefore classified as susceptibility (*S*) genes. There are many different kinds of *S*-genes, but generally they are up-regulated during plant–pathogen interactions, and they encode proteins that facilitate host colonization by the pathogen [106,132,133]. *S*-genes have emerged as a promising alternative to *R*-genes in PM resistance breeding, due to their broad-spectrum and potentially durable resistance characteristics [132,133,134]. There is a scope to search for *S*-gene orthologues in Pisum, using a comparative genomic approach. For example, fourteen *Mlo* genes were identified in *Medicago* [106]. With new genomic information in *Pisum*, additional *Mlo* loci can be identified. For such novel *S*-gene(s), loss of function mutations may be created by insertional mutagenesis [135], TILLING (Target induced local lesions in genomes) [136] and CRISPR/*Cas-9* (clustered regularly interspaced short palindromic repeats/CRISPR-associated protein9) [137] mediated genome editing approaches. This will ultimately result in the development of pea genotypes having wider PM resistance for more number of pathogen races [132]. Recently, the genome editing has been undertaken for the *Mlo* locus to achieve the resistance in different crops including wheat [138], grapevine [139], tomato [140] and sweet basil [141]. 

## 9. Host–Pathogen Interaction and Disease Development

*Pisum* species are diverse in their defense capacity against PM pathogens, and the genetic state of both pea plant and the *Erysiphe* spp. influences the outcome of the interaction. An intensive bidirectional signal exchange occurs between the plant and the PM fungus after spore deposition on the pea leaves until the late stages of the infection process [60]. When the pathogen overcomes the physical and chemical barriers present on the host, it identifies the pathogen associated molecular pattern (PAMPs) and/or effectors, thereby activating the PAMP-triggered immunity (PTI) and effector triggered immunity (ETI) [142]. Thus, in any susceptible genotype, the first step is suppression of PTI, which is achieved by the secretion of pathogen effectors, which manipulates the host cell function [143]. Therefore, for a better understanding of the pea–PM interaction during a compatible reaction, there is a need to find the way by which PTI is suppressed and ETS (effector triggered susceptibility) is established [144]. Pathogen effectors have the function of plant innate immunity suppression through host protein (effector target) interaction [143]. 

While studying the compatible and incompatible interaction between the pea and *E. pisi*, Bhosle et al. [75] identified several putative *er2* gene products having role in the expression of resistance. They identified the upregulation of transcripts of LRR receptor-like serine/threonine-protein kinase in the resistant cultivar suggesting its role in recognition and response to PAMPs [75]. The analysis of *E. pisi* lead to the identification of putative effectors such as GTPase, protein kinase, phosphatases, ATPase, DEAD box helicase, polyubiquitin, peptidyl-prolyl cis-trans isomerase, HSP70 and cytochrome P450. These effectors have a role in the pathogenesis and virulence [75]. A RPM-R homologue was found to upregulate in the resistant cultivar, suggesting a role in the recognition of the avirulence gene product of *E. pisi*. Similar RPM-R protein recognizes avrRpm1 type III effector avirulence protein in *Pseudomonas syringae* [145].

## 10. Conclusions

Pea productivity is constrained by PM disease, which is prevalent throughout pea growing regions. The concept of durable resistance in pea for PM resistance through gene pyramiding is still elusive, despite the idea of identifying more stable gene combinations across time and space being well established. In the absence of integrated and continuum germplasm enhancement and pre-breeding programmes, development of PM resistance at a host level is a long process, as it requires the use of wide genetic diversity from GP1/GP-II and GP-III. Among GP-1, cultivated resistant varieties, pre-bred lines (having *er2* and *Er3* gene), genetic and mutant stocks, and mapping populations developed in different pea breeding programs could be utilized (Figure 4). In order to discover novel PM-resistant genes/alleles, the germplasm resources of the GP-II and GP-III gene pool need to be explored systematically. The identified genetic resources could then be used to develop resistant cultivars by adopting the different breeding methods integrated with modern tools of genomics, MAS, transcriptomics, and genomic selection genome editing using CRISPR/Cas9. At the pathogen level, it is necessary to identify and characterize the virulent species/isolates across the different geographical regions of the world. The host–pathogen interaction (epidemiological studies) requires the attention of pathologist/breeders over the diverse growing habitat as the PM resistance is reportedly altered by environmental factors like temperature. A deep understanding of pathogen virulence genes and host resistance genes through transcriptomic and proteomics studies is still in its early stage.

## Figures and Tables

**Figure 1 genes-13-00316-f001:**
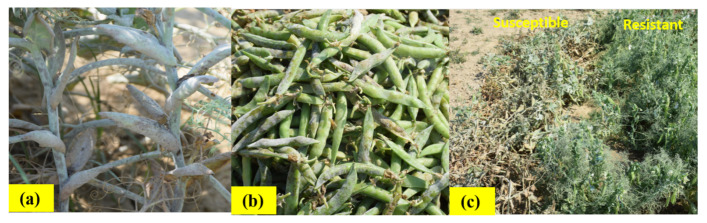
Powdery mildew in *Pisum sativum* L. (**a**): powdery growth of fungus on stems and leaves; (**b**): the affected pods of commercial cultivar ‘PC-531′ from India; (**c**): the susceptible and resistant lines growing under natural epiphytic conditions at ICAR-IIVR, Varanasi, India.

**Figure 2 genes-13-00316-f002:**
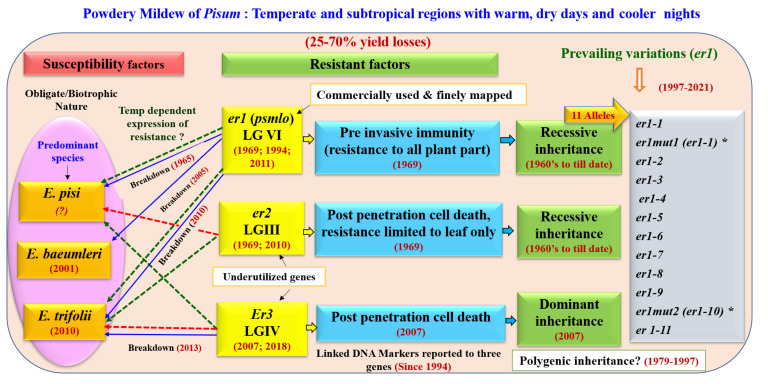
Summary of powdery mildew resistance in *Pisum* with the timeline of events. Among the three genes reported *er1* was harbored by many accessions and has now been characterized with 11 distinct alleles, of which *er1-1* and *er1-2* are currently used by the breeders. However, *er2* and *Er3* genes were reported in a few accessions only. Blue lines represent the breakdown of the *er1*- and *Er3*-mediated resistance by respective *Erysiphe* species. Green dotted lines denote the temperature-independent response of the resistant genes for respective species, while red dotted lines explain the temperature-dependent response (derived from information available in [20,21,22,30]); * represents the two induced mutation at *er1* locus where *er1mut1* is also designated as *er1-1; er1mut2* is also known as *er1-10*.

**Figure 3 genes-13-00316-f003:**
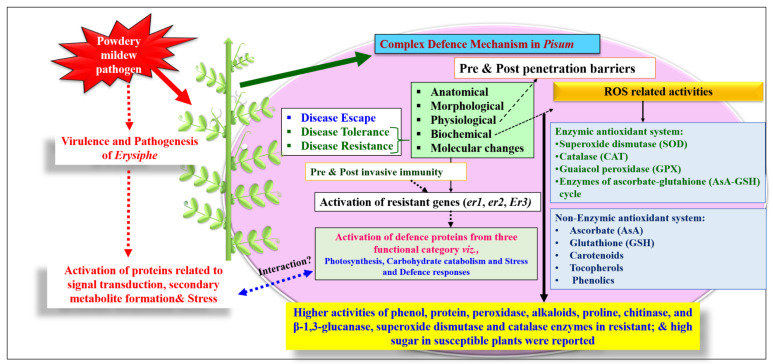
A comprehensive layout of plant defense mechanism seemingly operating in the peas. (Derived from [60,61,63,64,67,68,69]).

**Figure 4 genes-13-00316-f004:**
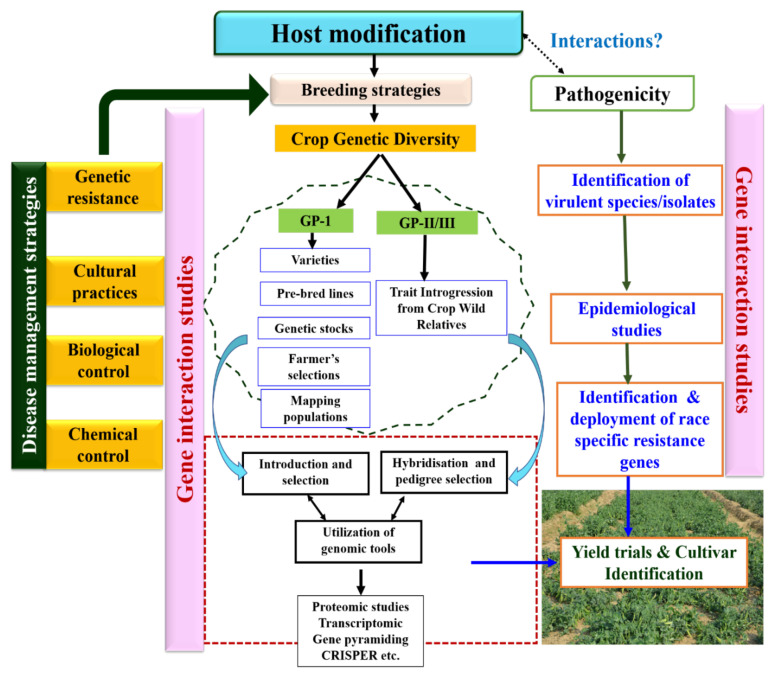
Scheme for powdery mildew management in peas, broadly advocating the utilization and characterization of pea genetic diversity along with the due emphasis on pathogen characterization for effective deployment of existing/novel variations reported for PMR.

**Table 1 genes-13-00316-t001:** Severity scores and corresponding proportions (%) of the surface area of leaves under disease and different scales used by several workers for powdery mildew scoring in pea.

Scale	Description (% Infection)	Marked as Resistant	Reference
0–9	0 = No infection 1 = 0.1–5%; 2 = 5.1–10%; 3 = 10.1–17%; 4 = 17.1–25%; 5 = 25.1–50%; 6 = 50.1–75%; 7 = 75.1–90%; 8 = 90.1–95%; 9 = 95.1–100%	0.1–10% = R; 10.1–30% = MR	[32]
0–10	0 = No infection; 1 = 5%; 2 = 10%; 3 = 15%; 4 = 40%; 5 = 33%; 6 = 46%; 7 = 60%; 8 = 73%; 9 = 86%; 10 = 100%	Not mentioned	[33]
0–9	1 = < 1%; 2 = 1–5%; 3 = 5–10%; 4 = 10–20%; 5 = 20–40%; 6 = 40–60%; 7 = 60–80%; 8 = 80–90%; 9 = > 90%	0–4 = R	[34]
1–5	1 = 1–5%; 2 = 6–20%; 3 = 21–30%; 4 = 31–75%; 5 = 76–100%	1–2 = R	[20]
0–4	0 = No mycelium growth; 1 = Sparse mycelium growth with little sporulation; 2 = Macroscopically: Slight mycelium growth; Microscopically: Slight to moderate mycelium growth with conidiophores; 3 = Macroscopically: Moderate mycelium growth; Microscopically: Moderate mycelium growth with moderate to heavy sporulation, 4 = Abundant mycelium growth and sporulation both micro- and macroscopically	0–2 = R	[23]
0–5	0 = No infection; 0.5 = < 10%; 1 = 11–20%; 1.5 = 21–30%; 2 = 31–40%; 2.5 = 41–50%; 3 = 51–60%; 3.5 = 61–70%; 4 = 71–80%; 4.5 = 81–90%; 5 = 91–100%	≤ 20% = R	[35]

Where R: Resistant; MR: Moderately Resistant.

**Table 2 genes-13-00316-t002:** Genetics of powdery mildew resistance in *Pisum*.

Cross	Generations	Genetics	Country	Reference
Unknown	F_2_	Cumulative factors for susceptibility	Sweden	[19]
Huancabamba × First of All	F_2_, F_3_	Single recessive gene	Peru	[36]
(B5115, B5243, B5064, B5806, PI2106613, PI280064, 46C, R300, NF, 477, 245, Early December, Satha, Bonneville, 31) × S-14	F_2_ and F_3_	Single recessive gene; duplicate recessive genes	India	[39]
Lincoln × (Wisconsin-7104, HPPC-63, HPPC-95, DPP-54, DPP-26 and S-143)	F_2_ and BCs, BC_R_	Single recessive gene	India	[43]
Radley × (JI 1559, JI 2480), JI 1758 × JI 2302, JI 1951 × JI 1648, JI 82 × JI 1648, Highlight × (JI 2302, JI 1559, JI 1210, JI 2480), JI 210 × JI 2302, JI 2480 × JI 1559	F_2,_ F_3_	Single recessive gene	Canada	[30,42]
P 1746 × MD 1-24, P 1744 × P 1760, P 1743 × HFP 4, HFP 4 × P1881, P 1744 × P1757, P 1742 × PG3, P 1746-8-1 × Pusa 10, P 1760 × Pusa 10, P 1746 × P 1746-1-1, P 1773-4 × P 1760	F_2_	Single recessive gene	India	[51]
M275-5-1 × Bohatyr, M275-5-1 × Jupiter, Green feast × M275-5-1, Traper × ATC1121, M275-5-1 × ATC1121	F_2,_ F_3_	Single recessive gene	Australia	[52]
C2 (*P. fulvum* line) × Messire	F_2,_ F_3_	Single dominant gene	Spain	[9]
Qizhen 76 × Xucai 1, Bawan 6 × Xucai 1, and Xucai 1 × Bawan 6	F_2_, and F_2:3_	Single recessive gene	China	[53]
Faloon × 11760-3^ER^	F_2_	Single recessive gene	Pakistan	[54]
Andina × ILS6527, San Isidro × ILS6527, Andina × UN6651, San Isidro × UN6651	F_2_, BCr, and BCs	Single recessive gene	Colombia	[35]
Stabil × i-6098881	F_2_	Single dominant gene	Russia	[37]

BCs (back cross susceptible); and BC_R_ (back-cross resistant): the country only represents the location of the experiment conducted and not the original source of these genotypes.

**Table 3 genes-13-00316-t003:** Temperature-based response of *er* genes along with their breakdown details.

Species	Response		Gene	
		*er1*	*er2*	*Er3*
*E. pisi*	Temperature response	Temp Independent [27]	Temp Dependent [27]	Temp Independent [9]
	Breakdown	Yes [22,42,56]	Yes [28]	Not Reported
*E. baeumleri*	Temperature response	Not Reported	Not Reported	Not Reported
	Breakdown	Yes [20]	Not Reported	Not Reported
*E. trifolii*	Temperature response	Temp Independent [22]	Temp independent [22]	Temperature dependent [22]
	Breakdown	Yes [20,21,22]	High resistant response [22]	Yes [22]

**Table 4 genes-13-00316-t004:** Characterization of *er1* gene and putative mutational events at *PsMLO1* locus.

*er1* Gene/Allele	Accession/Genotype	Mutational Event at *PsMLO1*	Reference
*er1-1*	JI 1559 (Mexique 4), Yunwan 8	C^680^G	[30,58,84]
*er1-1*	Tara and Cooper	-	[58,95]
*er1-1* (*er1 mut1*)	Induced mutation (Solara)	C/G transversion in exon 6	[47,89]
*er1-2*	JI 2302 (Stratagem)	Insertion of unknown size and identity	[30,84]
*er1-2*	G0006273 (X9002)	Insertion of unknown size and identity	[91,96]
*er1-2*	Xucai 1	129-bp deletion and 155-& 220-bp insertions	[94]
*er1-2*	Yunwan 21, Yunwan 23	-do-	[94]
*er1-2*	G0005576 (Wandou)	-do-	[59]
*er1-3*	JI210	ΔG at position 862 (exon 8)	[84]
*er1-4*	JI 1951/YI (landrace)	ΔA^91^ (frameshift)	[30,84]
*er1-5*	ROI3/02	G→A at position 570 (exon 5)	[85,92]
*er1-6*	G0001778 (landrace)	Point mutation (T→ C) at position 1121 (exon 11)	[59]
*er1-6*	G0002235	-do-	[91]
*er1-6*	G0002848	-do-	[91]
*er1-7*	DDR-11	10-bp deletion (TCATGTTATT) at exon 1 *(111-120)* of *PsMLO1*	[94]
*er1-7*	G0003895, G0003974	10-bp deletion (TCATGTTATT) at exon 1 of *PsMLO1*(111-120) and16-bp (CTCATCTTCCTCCAGG) deletion at position 776–792; and 16-bp (AATTTTTCTGTTTCAG) insertion at position 1171	[58]
*er1-7*	G0003931	10-bp deletion (TCATGTTATT) at exon 1 of *PsMLO1*(111-120) and 5-bp (GTTAG) deletion at position 700–704	[58]
*er1-7*	G0003936	-	[91]
*er1-7*	G0003899; G0003958 (DMR-26); G0003967	-	[91]
*er1-7*	G0004394	-	[91]
*er1-7*	G0003975	-	[91]
*er1-8*	G0004389	3-bp (GTG) deletion to positions 1339–1341 in exon 15	[91]
*er1-9*	G0004400	1-bp (T) deletion	[91]
*er1-10 (er1mut2)*	Induced mutation Frilene	G/A transition in exon 10	[47,89]
*er1 **** (er1-11)*	Yarrum and ps1771	2-bp insertion in intron 14	[93]

** The identified allele was not named in the studied population; however, later it was assigned as *er1-11* [90]; *er1mut1* is also known as *er1-1* [89,90]; *er1mut2* is also known as *er1-10* [90]; (−): Information not available.

**Table 5 genes-13-00316-t005:** Confirmed linkage groups of powdery mildew resistance genes in *Pisum*.

Gene	Location	Reference
*er1*	LGVI	[45,46,84,94,96,99,100,103]
*er2*	LGIII	[28,104]
*Er3*	LGIV	[102]

LGVI now assigned to the chromosome 1; LGIII to chromosome 5; and LGIV to chromosome 4 [1].

**Table 6 genes-13-00316-t006:** Powdery mildew-resistant *Pisum* accessions, gene diversity and screening details.

Immune/Resistant Accessions	Gene	Controlled Screening	Field Screening	Reference
SVP951, SVP952	*er2*	-	-	[55]
JI2480	*er2*	Yes	Yes	[28,40,62]
Highlight, AC Tamor, Tara, JI210, JI1951, JI82, JI1210, JI 2302	*er1*	Yes	Yes	[30,42]
Wisconsin-7104, HPPC-63, HPPC-95, DPP-26, DPP-54, S-143, Mexique-4, SVP-950, P6588	-	Yes	No	[43]
JP501A/2, NDVP-8, PMR-20	-	-	Yes	[110]
P1746, P1760, HFP4, P1442 (IC37255), P1746-8-1, P1779-4, P1746-24-1	*er* *	No	Yes	[51]
Glenroy, Kiley, Mukta, M257-3-6, M257-5-1, PSI11, ATC1181	*-*	No	Yes	[52]
Fallon, PS99102238, PS0010128	-	No	Yes	[31]
*er1mut1* (mutant from Solara), *er1mut2* (mutant from Frilene)	*er1*	Yes	Yes	[47]
Highlight, Mozart, AC Melfort, Fallon, Joell, Lifter, Franklin, Cebeco 1171, Tudor (Cebeco 4119), Cooper (Cebeco 1081), Lu 390—R2, SGL 1977, SGL 2024, SGL 444/2185, Carneval R, Consort R	*er1*	Yes	Yes	[20]
9057, 9370, 9375, 10609, 10612, 18293, 18412, 19598, 19611, 19616, 19727, 19750, 19782, 20126, 20152, 20171, It-96, No. 267, No. 380	-	-	-	[111]
IC208366, IC208378, IC218988, IC267142, IC278261	-	Yes	Yes	[23]
It-96, No. 267, JI2302	*er1*	Yes	Yes	[112]
Alaska, AC Tomour, Arka Ajit, Angoori, CHP-1 C-96, C-778, DAP-2, HUVP-3, JP-15, JP-20, JP-141, JP-625, Punjab -89, PMR-4, PMR-62, PMVAR-1, VRP-22, VRPMR-9, VRPMR-11, KTP-8; VP-233, JM-5, JP-501A/2, E-4, Vasundhra, JP-825	-	Yes	Yes	[29]
Arka Priya, Arka Pramod, Arka Ajit, IIHR 2-1, IPS-3	*er1*	No	Yes	[24]
KPMR-642, KPMR-516, KPMR-497, KPMR-557, VRPMR- 11	*er1*	Yes	Yes	[68]
HFPU, P-1797, P-1783, P-1052, HFP-7, HFP-8, P-1808, P-1820, P-1813, P-1377, P-1422-1, P-1811, IPF-99-25, KMNR-400, LFP-566, LFP-569, LFP-552, LFP-573, JP-501-A/2, PMR-21, KMNR-894, P-1280-4, P-1436-9, P-200-11, IPFD-99-13, HVDP-15, DPP-43-2, LFP-517, LFP-570, JP Ajjila, JP-15	-	Yes	Yes	[113]
Kashi Samridhi, VRPMR-10	*er1*	No	Yes	[2]
ILS6527, UN6651	*er1*	No	Yes	[35]
P660-4 (IFPI3261)	*Er3*	Yes	Yes	[9]
i-609881	** *Er3?*	Yes	-	[37]

* Resistance is governed by a single recessive gene. ** The resistance in i-609881 is single dominant gene, however, its identity as *Er3* is not yet confirmed.

**Table 7 genes-13-00316-t007:** The DNA markers linked to powdery mildew resistant genes.

Primer/Locus	Sequence	Distance (cM)	Marker	Gene	MP	Approach	References
*p236*	RFLP is restriction enzyme-based marker system	9.8	RFLP	*Er*	F_2_	-	[100]
*pI49*	RFLP is restriction enzyme-based marker system	18.0	RFLP	*er1*	RIL_S_	BSA	[99]
*pID18*	RFLP is restriction enzyme-based marker system	8.7	RFLP	*er1*	RIL_S_	BSA	[99]
PD 10	5′-GGTCTACACC-3′	2.1	RAPD	*er1*	RIL_S_	BSA	[99]
ScOPD10_650_ ^a^	(F) 5′-GGTCTACACCTCATATCTTGATGA-3′ (R) 5′-GGTCTACACCTAAACAGTGTCCGT-3′	2.1	SCAR	*er1*	RIL_S_	BSA	[99]
OPL-6	5′-GAGGGAAGAG-3′	2.0	RAPD	*er1*	F_3_	BSA	[114]
OPE-16	5′-GGTGACTGTG-3′	4.0	RAPD	*er1*	F_3_	BSA	[114]
*Sc-OPE-16_1600_ ^b^*	(F) 5′-GGTGACTGTGGAATGACAAA-3′ (R) 5′-GGTGACTGTGACAATTCCAG-3′	4.0	SCAR	*er1*	F_3_	BSA	[114]
* ^@^ * *Sc-OPO-18_1200_*	(F) 5′-CCCTCTCGCTATCCAATCC-3′ (R) 5′-CCTCTCGCTATCCGGTGTG-3′	-	SCAR	*er1*	F_3_	BSA	[114]
OPO-02	5′-ACGTAGCGTC-3′	4.5	RAPD	*er1*	NILs	-	[45]
OPU-17	5′-ACCTGGGGAG-3′	10.3	RAPD	*er1*	NILs	-	[45]
ScOPD 10_650_ ^a^	(F) 5′-GGTCTACACCTCATATCTTGATGA-3′ (R) 5′-GGTCTACACCTAAACAGTGTCCGT-3′	3.4	SCAR	*er1*	NILs	-	[45]
A5 ^c^	(F) 5′-GTAAAGCATAAGGGGATTCTCAT-3′ (R) 5′-CAGCTTTTAACTCATCTGACACA-3′	20.9	SSR	*er1*	F_2_	NA	[115]
PSMPSAD60 ^d^	(F) 5′-CTGAAGCACTTTTGACAACTAC-3′ (R) 5′-ATCATATAGCGACGAATACACC-3′	10.4	SSR	*er1*	F_2_	BSA	[46]
PSMPSAA374e	(F) 5′-GTCAATATCTCCAATGGTAACG-3′ (R) 5′-GCATTTGTGTAGTTGTAATTTCAT-3′	11.6	SSR	*er1*	F_2_	BSA	[46]
PSMPA5 ^c^	(F) 5′-GTAAAGCATAAGGGGATTCTCAT-3′ (R) 5′-CAGCTTTTAACTCATCTGACACA-3′	14.9	SSR	*er1*	F_2_	BSA	[46]
PSMPSAA369	(F) 5′-CCCTTCGCACACCATTCTA-3′ (R) 5′-AGTCGTTTTGGAGATCTGTTCA-3′	24.1	SSR	*er1*	F_2_	BSA	[46]
PSMPSAD51	(F) 5′-ATGAAGTAGGCATAGCGAAGAT-3′ (R) 5′-GATTAAATAAAGTTCGATGGCG-3′	25.8	SSR	*er1*	F_2_	BSA	[46]
OPWO4_637	5′-CAGAAGCGGA-3′	-	RAPD	*Er3*	F_2_	BSA	[101]
OPAB01_874	5′-CCGTCGGTAGT-3′	2.8	RAPD	*Er3*	F_2_	BSA	[101]
*SCAB1 _874_*	(F) 5′-CCGTCGGTAGTAAAAAAAACTA-3′ (R) 5′-CCGTCGGTAGCCACACCA-3′	2.8	SCAR	*Er3*	F_2_	BSA	[101]
ScW4_637_	(F) 5′-CAGAAGCGGATGAGGCGGA-3′ (R) 5′-CAGAAGCGGATACAGTACTAAC-3′	-	SCAR	*Er3*	F_2_	BSA	[101]
*ScX17_1400_*	(F) 5′-GGACCAAGCTCG GATCTTTC-3′ (R) 5′-GACACG GACCCAATGACATC-3′	2.6	SCAR	*er2*	F_2_	BSA	[28]
ScOPO06_1100_y	(F) 5′-CCCCATGTTAGAACCTTGCA-3′ (R) 5′-ACGGGAAGGTCTGACAGTAT-3′	0.5	SCAR	*er1*	NILs	BSA	[116]
ScOPT16_480_	(F) 5′-GGGCAGAATCAGCTGAGCTC-3′ (R) 5′-GAACAAGGAGAAGAAGAGG-3′	3.3	SCAR	*er1*	NILs	BSA	[116]
ScAGG/CAA_125_	(F) 5′-GAATTCAGGAACATAGCTTC-3′ (R) 5′-CAAGCTAAAAGTCAGAAGAT-3′	5.5	SCAR	*er1*	NILs	BSA	[116]
ScOPE16 ^b^	(F) 5′-GGTGACTGTGGAATGACAAA-3′ (R) 5′-GGTGACTGTGACAATTCCAG-3′	9.2	SCAR	*er1*	NILs	BSA	[116]
A5 ^c^	(F) 5′-GTAAAGCATAAGGGGATTCTCAT-3′ (R) 5′-CAGCTTTTAACTCATCTGACACA-3′	23.0	SSR	*er1*	NILs	BSA	[116]
BC210	-	8.2	RAPD/SCAR	*er1*	-	-	[103]
OPB18_430_	5′-CCACAGCAGT-3′	11.2	RAPD	*er1*	F_2_	-	[54]
ScOPX04_880_	(F) 5′-CCGCTACCGATGTTATGTTTG-3′ (R) 5′-CCGCTACCGAACTGGTT GGA-3′	0.6	SCAR	*er1*	NILs	BSA	[117]
ScOPD 10_650_ ^a^	(F) 5′-GGTCTACACCTCATATCTTGATGA-3′ (R) 5′-GGTCTACACCTAAACAGTGTCCGT-3′	2.2	SCAR	*er1*	NILs	BSA	[117]
AD60 ^d^	(F) 5′-CTGAAGCACTTTTGACAACTAC-3′ (R) 5′-ATCATATAGCGACGAATACACC-3′	9.9 *, 8.7 **	SSR	*er1*	F_2_	BSA	[53]
c5DNAmet	(F) 5′-TTCTTACTGTTCGTGAATGCGCC-3′ (R) 5′-GCCCTAATCCTCTAATTGGCGCTC-3′	15.4 *, 8.1 **	SSR	*er1*	F_2_	BSA	[53]
AD61	(F) 5′-CTCATTCAATGATGATAATCCTA-3′ (R) 5′-ATGAGGTACTTGTGTGAGATAAA-3′	0.39	SSR	*Er3*	F_2_	BSA	[102]

Where a, b, c, d denotes the same primer used by different researchers; @ This fragment was only present in susceptible progenies; *—in mapping population ‘Xucai 1 × Bawan 6′; **—in mapping population ‘Qizhen 76 × Xucai 1’; Information for the marker BC210 is not available. Where, RFLP: Restriction Fragment Length Polymorphism; RAPD: Random Amplified Polymorphic DNA; SCAR: Sequence Characterized Amplified Region; SSR: Simple Sequence Repeat; RILs: Recombinant Inbred Lines; NILs: Near-Isogenic Lines; BSA: Bulked Segregant Analysis.

**Table 8 genes-13-00316-t008:** Allelic diversity studies in *Pisum sativum* with the available details.

Accessions	Disease Score	PM Isolate	Genetics (Gene)	Mapping Population (Generation)	Nearest Marker (Linkage Distance in cM)	Reference
C2 (P660-4, *P. fulvum*)	R*	CO-01	*SDG (Er3)*	C2 × Messire (F_2_ & F_2_:_3_)	SCAB1_874_ (2.8 cM)	[9,101]
Eritreo (breeding line C2)	R*	NP	*SDG (Er3)*	C2 × Messire (F_2_)	AD61 (0.39 cM)	[102]
Xucai 1	R*	EPBJ	*SRG (er1*-2)	Xucai1 × Bawan6 (F_2_);	AD60 (9.9 cM) and c5DNAmet (15.4)	[53]
Xucai 1	R*	EPBJ	*SRG (er1*-2)	Qizhen76 × Xucai1 (F_2_)	AD60 (8.7 cM) and c5DNAmet (8.1 cM)	[53]
G0006273 (X9002)	I (0)	EPYN	*SRG (er1-2)*	Bawan 6 × X9002 (F_2_)	AD60 (11.9 cM), c5DNAmet (9.0 cM); PsMLO1-650 (FM)	[96]
G0001778 (Dabaiwandou), G0001752, G0001763; G0001764; G0001767, G0001768; G0001777; G0001778; G0001780; G0003824; G0003825 and G0003826	I (0)	EPYN	*SRG (er1-6)*	G0001778 × Bawan 6 (F_2_ and F_2_:_3_)	SNP1121 (FM); AD60 (8.8 cM) and c5DNAmet (22.8 cM)	[59]
DDR-11	I (0)	EPYN	*SRG (er1-7)*	DDR-11 × Bawan (F_2_ & F_2_:_3_)	ScOPD10-650 (8.3 cM) PSMPSAD60 (4.2 cM); ScOPE16-1600 (21.4 cM); PSMPSA5 (9.5 cM); c5DNAmet (26.2 cM)	[58]
G0004389	I (0)	EPYN	*SRG (er1-8)*	WSU 28 × G0004389 (F_2_ & F_2_:_3_)	c5DNAmet (9.6 cM); AA200 (3.5)	[91]
G0004400	I (0)	EPYN	*SRG (er1-9)*	Bawan6 × G0004400 (F_2_ & F_2_:_3_)	PSMPSAD51 (12.2 cM); ScOPX04-880 (4.2 cM)	[91]
Yarrum and ps1771	R*	NM	*SRG (er1-11)*	Kaspa × Yarrum; Kaspa × ps1771 (RIL)	AB71 (4.6 cM) and AD59 (4.3 cM)	[93,115]

Where I, R and S indicate resistance levels *viz*., immune, resistant, susceptible; R*: Disease score not mentioned; SRG Single recessive gene, SDG: Single dominant gene; EPBJ (NCBI accession number KR912079); EPYN (NCBI, accession number KR957355; FM: functional marker.

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
