# Peer review of "Gene-Based Resistance to Erysiphe Species Causing Powdery Mildew Disease in Peas (Pisum sativum L.)"

_genes, 2022, doi:10.3390/genes13020316_

Round 1

Reviewer 1 Report

The manuscript entitle of "Advancements in er/Er gene-based Resistance to Erysiphe species Causing Powdery Mildew Disease in Peas (Pisum sativum L.)" sounded in science and technology. The value of this review will be used for related field in future.

However, the author should be revise some point for made the review manuscript is suitable.

The general comment are;

  1. Page 1, Authors information – please add some details about author’s information and E-mail.
  2. Lines 63-70 should be citation some references.
  3. The seasonal variation or GxE interaction evaluation of PM in pea should be added in some where if had been reported.
  4. The Figure 1C, the resistance line are not appeared in figure (Please check).
  5. The figure 2 should be mention in Topic 4.2 (the figure should be mention somewhere before it appeared).
  6. The front in Figure should be format with the Journal style.
  7. Figure 3, the citation were shown in figure, it should be as journal format?
  8. The text is not correctly formatted. Most tables that are horizontal should be placed vertically.
  9. Topic 6.2, please mention more about Linkage map / QTL mapping and the comparative linkage map of PM resistance in pea.
  10. Topic 7.1, please provided the information of Er3 in Table 6 too. Due to lines 183-186 the author was mention to Er3.
  11. The information of topic 7.2 (lines 371-400), which limited to the gene pyramid of those 3 er genes (er1, er2 and Er3)? Please conclude.
  12. The figure 4, the Lathyrus and Vicia species are needed to add for somewhere in the figure?
  13. The information of reciprocal cross (lines 446-448) the results indicated?
  14. The end of topic 8.1, please conclude the concept of gene transfer from pea gene pool.
  15. Lines 456-457 are correctly? The information in table to indicate by genetic control.
  16. Topic 8.3, base on the information of topic 7.2 (lines 381-384), the er genes can be pyramid or not?
  17. Topic 8.5, Can the author conclude or discussion why the gene pyramid of er gene still not yet now?
  18. Some references (5, 8, 13, ect) are not journal format. Please check all.

Reviewer 2 Report

In this manuscript authors reviewed the role of er/Er genes in Pea's resistance against Erysiphe. This information is useful as many information are provided for researchers to study and improve the resistance cultivar of Pea against the fungi, as well as study of fungal pathogenesis.

However it is not seen about molecular interaction of host-pathogen response between those er/Er genes in pea - avirulent genes of Erysiphe in PTI and ETI interaction. This information will give more comprehensive and useful information in study of pea resistance mechanism.

Authors may wish to address the following point arisen in reviewing this manuscript.

  1. some literatures needs to be cited: line 70; line 134; line 162 (158 -162); line 254; line 400.
  2. Figure 1.c. It is written: susceptible and resistant lines... but the resistant lines were not shown in the picture. Please elaborate.  
  3. Figure 2. It is written Er2 but in the text and legend are written er2 (recessive gene). Please confirm.
  4. Line 199-209 relates to Table 3. Authors wrote temperature-based response of the genes are different in 3 Erysiphe spp., but there is not enough review and discussion what influence their different response. Please elaborate further for this information, what influence the temperature-based response is different among those genes and those species of Erysiphe.
  5. Line 235; Chapter 5. It will be more informative if authors add the molecular aspect as well, therefore become: Biochemical and Molecular Interaction for PM Resistance. It could be divided into 2 sub-chapter 5.1. The Biochemical aspect; 5.2. The molecular aspect. In this sub-chapter please review the avirulent genes of PM pathogens in which er1, er2 and ER3 are responsible for.
  6. Line 285-287: Please add why HSPs plays important role in survival and spread of biotrophic Erysiphe. How about in the pathogenesis?
  7. Table 7. Why not all information in the table has references?
  8. Line 496. Sub-chapter 8.4. Why there is no explanation about what kind of S genes on pea plants against PM? Please add any update of transcriptomic research relates to S genes of Pea against PM? 

Thank you very much. 

Round 2

Reviewer 1 Report

Dear authors,

Thanks for incorporating my feedback. The manuscript generally looks good on my end. I appreciate your revision, and after your major revision on manuscript, I decided that your manuscript is accepted.

Thank you.

Author Response

Added as editors response

Reviewer 2 Report

Dear authors,

I appreciate your revision, and after your major revision on manuscript, I decided that your manuscript is accepted. 

Thank you. 

Author Response

Added as the editors response